# Thermal transport mechanisms in ZIFs

Xiaoqi Zhang ®[1], Senja Barthel ®[2], Yutao Li ®[1], Berend Smit ®[1]✉ & Raffaela Cabriolu ®[3]✉

Zeolitic imidazolate frameworks (ZIFs), a subclass of metal-organic frameworks (MOFs), exhibit tunable thermal conductivity, which is crucial for applications such as gas adsorption, catalysis, and energy storage. Despite its significance, thermal conductivity in MOFs remains less explored compared to other key performance indicators. In this work, we investigate the thermal transport properties of 196 ZIF structures with diverse net topologies and organic linkers. We develop a thermal circuit model that quantitatively integrates network topology and consolidates various atomic contributions into heat conduction units for thermal analysis. Our results reveal a strong correlation between circuit-estimated and simulated thermal conductivity, demonstrating the model's predictive power. Additionally, we find that functional groups influence thermal transport through a competing interplay between atomic mass and mechanical stability. These findings provide a systematic approach for tailoring MOF thermal properties, offering insights into the rational design of materials with optimized thermal performance.

Metal-organic frameworks (MOFs) are a class of crystalline materials composed of metal clusters and organic ligands. Their structural tunability and high porosity make them promising for various applications[1] such as gas adsorption and separation[2–4], catalysis[5,6], and drug delivery[7–9]. Researchers have developed various key performance indicators (KPIs) to assess MOFs for specific applications and have extensively studied their potential[10,11]. New materials are designed using advanced machine-learning techniques based on these KPIs[12]. However, thermal conductivity is often overlooked despite its potential to impact the MOFs' ranking. For example, poor heat dissipation within the structure can reduce the amount of gas adsorbed[13–15]. Similarly, the rate of heat conduction in MOFs can affect reaction rates and selectivity in catalytic processes[16].

Recent studies have increasingly focused on the thermal conductivity of MOFs. Islamov et al.[17] conducted high-throughput molecular simulations on 10194 hypothetical MOFs generated by ToBaCCo, revealing that high thermal conductivity is associated with high density, small pores, and four-coordinated metal nodes. Similarly, Lin et al.[18] found that close metal atom arrangements and a high proportion of metal and transition metal atoms contribute to higher thermal conductivity in their simulations of 1214 structures from the CoRE MOF database. Comparable correlations between thermal conductivity and geometric characteristics, such as pore size[19–21] and density[22], have also been extensively investigated in covalent organic frameworks (COFs). Due to the more uniform covalent bonding environments in COFs, these structure-property relationships generally exhibit stronger consistency. While these high-throughput simulations[18,21] have identified some common trends, such as the influence of pore size and metal connections[17,18], other factors like mass mismatch[17,23], and the impact of functional groups[24], the net topology[25], and pore shape[26] remain less understood.

Zeolitic imidazolate frameworks (ZIFs) are a distinct subclass of MOFs, consisting of tetrahedrally coordinated transition metal ions linked by imidazolate (IM) ligands. Ying et al.[24] studied ZIF-8 with four different functional groups and found that thermal conductivity $\kappa$ followed the trend: $\kappa$(ZIF-8(CH$_3$)) > $\kappa$(ZIF-8(H)) > $\kappa$(ZIF-8(Br)) > $\kappa$(ZIF-8(Cl)). To gain broader insights into the structure-property relationship, Cheng et al.[25] investigated 18 ZIFs with the same organic ligands (2-methyl imidazolate (mIM)) but varying net topologies. They introduced two parameters: alignment tensor and pathway factor, to quantitatively evaluate the orientation and distribution of heat transfer pathways within

[1]Laboratory of Molecular Simulation (LSMO), Institut des Sciences et Ingénierie Chimiques, École Polytechnique Fédérale de Lausanne (EPFL), Sion, Switzerland. [2]Department of Mathematics, Vrije Universiteit Amsterdam, Amsterdam, the Netherlands. [3]Department of Physics, Norwegian University of Science and Technology (NTNU), Trondheim, Norway. ✉e-mail: berend.smit@epfl.ch; raffaela.cabriolu@ntnu.no

frameworks. However, these parameters are not easily generalizable to a broader range of MOFs[18].

In MOFs, we typically have three parameters we can tune: the metal node, linker, and network topology. In this work, we focus on both the network topology and functionalization of the linker. Experimentally, for ZIFs only, we have been able to obtain a large number of ZIFs with the same metal node and linker but with different network topologies[27]. We expanded the understanding of thermal transport in ZIFs by simulating 196 structures with various net topologies and organic linkers. We developed a thermal circuit model to analyze the net topology of a structure in heat conduction quantitatively. Furthermore, we highlight how the mechanical stability changes and mass variations introduced by functional groups jointly influence thermal transport. In principle, similar studies can be done with MOFs, but there are far fewer experimental examples in which we can synthesize MOFs with the same linker and metal node but different pore topologies[28].

## Results

### Thermal conductivity of ZIFs

The ZIFs studied in this work are the same set that was used by Moosavi et al.[29] to investigate the mechanical properties. To ensure the chemical consistency of all 196 ZIFs, we have run `mofchecker`[30], and no issues were detected. We performed equilibrium molecular dynamics (MD) simulations for 196 ZIFs with 49 different net topologies and four different organic ligands. The organic ligands include IM, dichloro imidazolate (dcIM), mIM, and 2-nitro imidazolate (nIM), whose structures are shown in Fig. 1a. Each zinc atom is tetrahedrally coordinated with four organic ligands. The results of these simulations are summarized in Fig. 1b, c. The numerical results can be found on Zenodo[31].

Previous studies have identified a linear relationship between thermal conductivity $\kappa$ and geometric properties such as density[13,25] and pore size[25,26]. More recently, Islamov et al.[17] expanded on this finding by demonstrating a general trend between density and thermal conductivity through simulations of a large, diverse MOF dataset.

Figure 1b shows that for the ZIFs we studied, the thermal conductivity follows a linear relationship against density for each metal-ligand combination (distinguished by color). However, no overall correlation is found when considering all data points together (Pearson coefficient $r = -0.37$). An increase in density does not always correspond to enhanced thermal conductivity. A similar analysis of the correlation between void fraction and thermal conductivity can be found in Section 1 of Supplementary Information (SI).

Figure 1c shows thermal conductivity as a function of volumetric atom density, resulting in an improved overall Pearson coefficient of 0.78. Despite this improvement compared to density, volumetric atom density may not be an ideal indicator. Indeed, Fig. 1c suggests that atoms in the backbone and those in functional groups (e.g., Cl in dcIM, C and H in mIM, and N and O in nIM) likely contribute differently to thermal conductivity.

The unsatisfactory correlations between density, volumetric atom density, and thermal conductivity can be attributed to two main factors. First, from our studies, the contribution of each atomic species to thermal conductivity depends on its mass and chemical environment. Density and volumetric atom density fail to capture the varying effects. Second, these simple density-based correlations do not explicitly account for the network and connectivity within the structures. These observations motivated the search for alternative indicators that consider these factors, with the potential to explain thermal conductivity more effectively.

### Thermal circuit model

Heat conduction is analogous to electrical conduction: the former is driven by temperature gradients, while potential gradients drive the latter. This analogy has been exploited to study thermal transport in continuum materials[32,33]. However, its application in simulating heat transfer within molecules remains unexplored. By transforming a crystal network into a thermal circuit network, we calculate thermal conductivity using an analogy to electrical conductivity. In the remainder, we refer to this quantity as "thermal circuit conductivity".

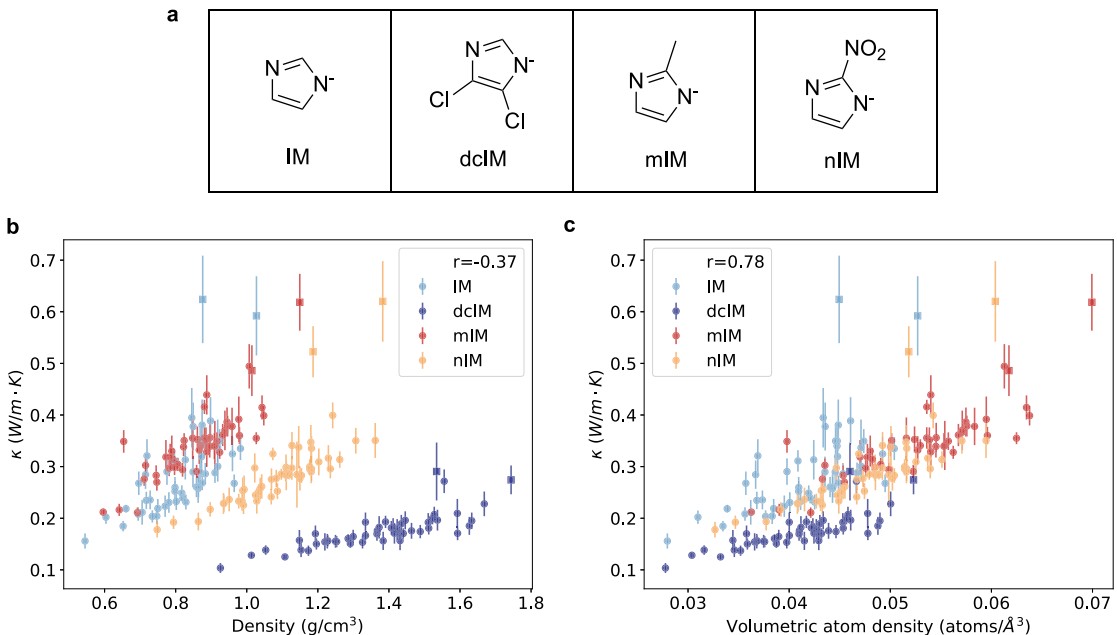

**Fig. 1 | Structures of the four organic ligands used in this study and their corresponding MD thermal conductivities. a** Molecular structures of the organic ligands: imidazolate (IM), 2-methyl imidazolate (mIM), 2-nitro imidazolate (nIM), and dichloro imidazolate (dcIM). **b** Correlation between thermal conductivity and density. **c** Correlation between thermal conductivity and volumetric atom density.

A linear relationship between density and thermal conductivity is observed for each specific combination of metal nodes and organic ligands. Two exceptional net topologies, BCT and SOD, are represented by squares in (**b**) and (**c**). The Pearson coefficients $r$ are computed excluding structures with these two net topologies, and the errors are calculated according to the procedure in Section 2 of the SI.

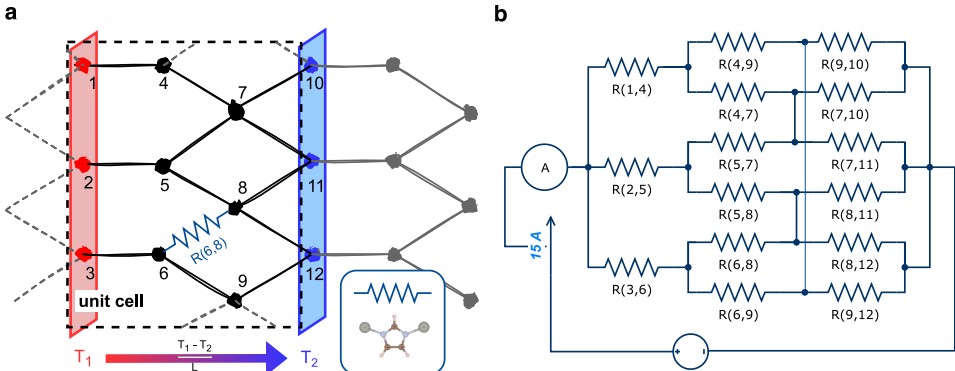

**Fig. 2 | An example of a 2D crystal network and its corresponding electrical circuit. a** Nodes represent atoms at the intersections of heat conduction paths, and edges denote the connecting atoms and chemical bonds. In ZIFs, each node corresponds to a Zn atom, and each edge to a Zn-linker-Zn "heat conduction unit"

(inset). **b** Each resistor, with a resistance of $1\,\Omega$, represents a heat conduction unit. A voltage of 1 V is applied to the circuit, and the current flowing through it is computed.

An example of a two-dimensional periodic crystal network is shown in Fig. 2a. In this representation, nodes correspond to atoms (or groups of atoms) located at the intersections of heat conduction paths, while edges denote the connecting atoms and chemical bonds. A temperature difference, e.g., $T_1$ on the left and $T_2$ on the right side of a unit cell, drives the heat flux. By simulating the resulting heat flow, the thermal conductivity of the crystal network can be determined.

To model this, we use an electricity analogy, mapping the crystal structure to a circuit. On the left boundary of the unit cell, the nodes with bonds crossing the left unit cell boundary are selected as starting points, and their equivalent nodes in the adjacent unit cell serve as endpoints. These starting and endpoint nodes are treated as equipotential. Applying a voltage $U$ across them creates a uniform electric field $E_i = U/L_i$, corresponding to a constant potential gradient along the unit cell direction $L_i$. The current density is given by $J_i = I/A_{\perp i}$, where $A_{\perp i}$ is the cross-sectional area perpendicular to the transport direction. Figure 2b shows the circuit corresponding to the two-dimensional (2D) crystal network. In this illustrative case, with $U = 1$ V and each resistor set to a resistance of $R = 1\,\Omega$, the total current is $I = 15$ A. In the general three-dimensional case, Ohm's law yields the circuit conductivity as

$$\sigma_i = \frac{J_i}{E_i} = \frac{I/A_{\perp i}}{U/L_i}, i = x, y, z. \tag{1}$$

In the study of ZIFs, we construct the circuit model by defining the zinc atoms as the network nodes. A resistor is placed between two nodes when they are connected by an organic linker, representing a heat conduction unit (Zn-linker-Zn, where the linker is IM, dcIM, mIM, or nIM), as shown in the inset of Fig. 2a. The corresponding circuit is generated using an in-house Python code[31].

Since the resistance of an individual heat conduction unit is not known a priori, we employ a reduced (dimensionless) formulation. Specifically, we set $U^* = 1$ and assign a resistance of unity to each heat conduction unit. The reduced current $I^*$ is then obtained by circuit simulation in MATLAB Simulink[34]. The reduced circuit conductivity $\sigma_i^*$ is calculated as

$$\sigma_i^* = \frac{I^*}{U^*} \cdot \frac{L_i^*}{A_{\perp i}^*}, \quad i = x, y, z, \tag{2}$$

where $A_{\perp i}^*$ is the reduced cross-sectional area of the structure, $L_i^*$ is the reduced lattice length in the computed direction. The reduced thermal circuit conductivity $\tau_i^*$ equals the value of $\sigma_i^*$. Equation (2) gives us the equivalent reduced circuit conductivity for each lattice direction by constructing the circuit along the respective direction. The overall reduced thermal circuit conductivity is calculated as the average

conductivity across $x$, $y$, and $z$ directions. The correlation between reduced thermal circuit conductivity $\tau^*$ and MD thermal conductivity $\kappa$ is shown in Fig. S7 in the SI.

Once we have a set of data for the same type of heat conduction units, namely, reduced thermal circuit conductivity ($\tau^*$) and MD thermal conductivity ($\kappa$), we can statistically estimate the thermal resistance of the heat conduction unit. Specifically, we fitted a linear model between the reduced thermal circuit conductivity and the MD thermal conductivity for each group of ZIFs with the same linker. Using the linker-specific scaling factors obtained from the fitting, we can convert the reduced thermal circuit conductivity ($\tau^*$) to a thermal circuit conductivity ($\tau$) in real units. Details about the fitting process and the scaling factors can be found in Section 7 of SI.

With these scaling factors, the thermal circuit conductivities ($\tau$) of the four types of ZIFs converge onto a single universal linear correlation with the MD thermal conductivity. The thermal circuit conductivity exhibits a strong and universal correlation with the MD thermal conductivity, with a Pearson correlation coefficient of 0.92, as shown in Fig. 3. By treating atoms in different chemical environments as collective heat conduction units, the circuit model accounts for varying atomic contributions to thermal transport. This approach enhances the overall correlation ($r = 0.92$) compared to that with density ($r = -0.37$) and volumetric atom density ($r = 0.78$).

### Effect of functional groups

Figure 1b shows that ZIFs with the same net topology but heavier atoms can exhibit higher density yet lower thermal conductivity. For example, replacing imidazolate with 2-nitro imidazolate increases the framework density but lowers the heat conductivity. In contrast, volumetric atom density cannot distinguish ZIFs with different thermal conductivity. For instance, ZIFs(dcIM) exhibit lower thermal conductivity despite having a similar volumetric atom density to ZIFs(IM). By considering collective atoms as a heat conduction unit, the true effect of adding a functional group becomes evident in Fig. 3. Functionalization has minimal impact on ZIFs(nIM), enhances thermal conductivity in ZIFs(mIM), and reduces it in ZIFs(dcIM). This trend, $\kappa_{mIM} > \kappa_{nIM} \approx \kappa_{IM} > \kappa_{dcIM}$, is also reflected in the fitted scaling factors (see Table S2 in the SI).

If we add a functional group, two things can happen. First, the mass of the ligand changes, and it is well-known that differences in mass can have a large effect on thermal conductivity. A second effect is that a functional group can impact the material's stiffness[29]. Structural stiffness and thermal conductivity are inherently linked, as strong bonds suppress atomic vibrations and enhance phonon transport, leading to both high mechanical stability and increased thermal conductivity.

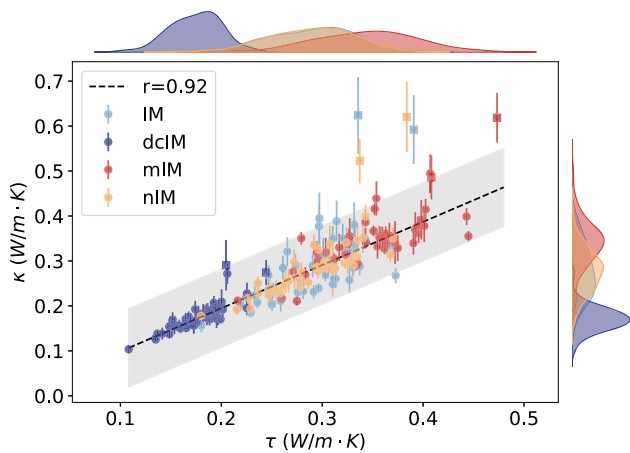

**Fig. 3 | The MD thermal conductivity $\kappa$ versus the thermal circuit conductivity $\tau$.** Histograms of $\tau$ and $\kappa$, distinguished by linker type, are shown along the x- and y-axes, respectively. The scaling factors are fitted for ZIFs with identical linkers, yielding good linear correlations when a single coefficient is applied. Pearson coefficients are calculated after excluding two exceptional net topologies, BCT and SOD, which are represented by squares. The gray shading indicates measurement uncertainty, estimated by repeating the current measurement three times with different choices of the unit cell.

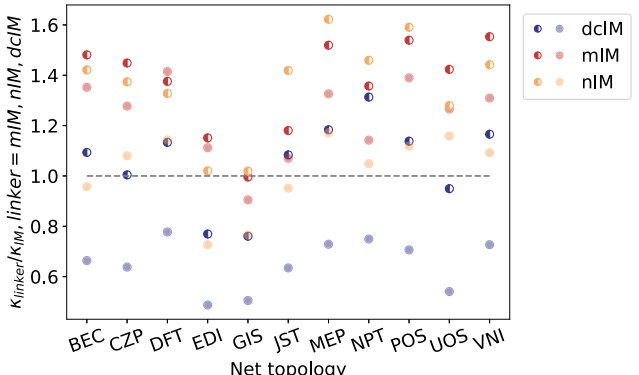

**Fig. 4 | Thermal conductivity ratio of ZIFs with functionalized linkers (mIM, nIM, and dcIM) relative to ZIFs(IM).** Half-filled markers represent cases where linker masses are artificially scaled to match that of the lightest linker (IM), which shows a similar trend to mechanical stability. Solid circles indicate results under original mass conditions. The heaviest linker, dcIM, leads to the greatest reduction in thermal conductivity, whereas mIM, with the smallest mass difference from IM, results in only a modest reduction.

To investigate these effects, we first carry out a simulation in which we remove the mass difference by modifying the ligands' atomic masses so that all four organic linkers have the same mass as the lightest linker, IM. This is not a simulation that mimics an experiment one can do in the real world. Eleven net topologies were randomly selected for these simulations. The results, shown as half-filled markers in Fig. 4, reveal a distinct trend: when the mass of the linker is the same, the introduction of functional groups increases thermal conductivity compared to ZIFs (IM). This effect is most pronounced in ZIFs (mIM) and ZIFs (nIM), where the thermal conductivity ratio exceeds 1, whereas ZIFs (dcIM) exhibit a more modest increase.

Now that we have (artificially) eliminated the effect of mass, the remainder of the effect can be explained in terms of changes in the rigidity of the network. Indeed, previous work on the mechanical stability of MOFs has highlighted a similar impact of functional groups on network stiffness. Moosavi et al.[29] demonstrated that strategically placed functional groups can enhance mechanical stability by forming a secondary network of nonbonded interactions. Figure S3 in the SI, which is based on data from ref. [29], shows that ZIFs with mIM and nIM linkers have a higher bulk modulus, and thus greater mechanical stability, compared to those with IM and dcIM linkers.

In Section 5 in the SI, we further show that ZIFs with functional groups exhibit a higher contribution of van der Waals interactions to total thermal conductivity than their IM-linked counterparts. These results support the idea that enhanced mechanical stability, driven by non-bonded interactions, increases thermal conductivity.

Furthermore, our calculations reveal that mass plays a non-negligible role in heat conduction. The heaviest linker, dcIM, results in the greatest reduction in thermal conductivity, as evidenced by the large difference between the half-filled and solid markers for dcIM in Fig. 4. In contrast, for nIM, the competing effects of mass and mechanical stability approximately cancel each other out. In ZIFs(mIM), the dominant influence of enhanced mechanical stability leads to the highest observed thermal conductivity. This interplay between mass and mechanical stability leads to the trend: $\kappa_{mIM} > \kappa_{nIM} \approx \kappa_{IM} > \kappa_{dcIM}$.

Two exceptions are observed in the network analysis: BCT in the z-direction and isotropic SOD in x-, y-, and z-directions, which can be attributed to their relatively high bulk modulus[29] and similar zigzag heat conduction pathways (see Section 6 in the SI for more details). In addition, these two topologies are characterized by the presence of 4-membered and 6-membered rings, structural motifs that tend to enhance the vibrational freedom of the metal centers[35,36]. These additional vibrations may lead to variations in the thermal resistance of the corresponding heat conduction units.

## Discussion

In this study, we investigated the thermal conductivity of ZIFs by simulating 196 structures, covering 49 distinct net topologies and four organic ligands. We propose a thermal circuit model that allows us to estimate "thermal circuit conductivity" as an analog to thermal conductivity. This approach effectively addresses two key limitations of traditional density-based indicators: (1) it collectively accounts for the varying contributions of atoms with different masses and chemical environments to thermal conductivity, and (2) it incorporates the crystal network in heat conduction analysis. Our results demonstrate a strong correlation between circuit-estimated and MD thermal conductivity, providing a reliable tool for predicting thermal transport in ZIFs. Additionally, we found that the interplay between increased mass and enhanced mechanical stability, introduced by functional groups, significantly impact on thermal conductivity. While increased mass reduces thermal conductivity, greater mechanical stability facilitates heat transport. Although our work focuses on ZIFs, the proposed model provides a generalizable and efficient strategy that holds strong potential for extension to a broader range of MOFs, offering guidance for tailoring thermal conductivity through topology design and functionalization.

## Methods

We performed MD simulations using the Large-scale Atomic/Molecular Massively Parallel Simulator (LAMMPS) software[37]. The heat flux was computed using the algorithm of Boone et al.[38]. The input files were generated using the lammps-interface package[39]. The frameworks were equilibrated with the isothermal-isobaric (NPT) ensemble at 300 K and 1 bar for 10 ns, followed by a canonical (NVT) ensemble at 300 K for another 10 ns. The Nosé-Hoover thermostat and barostat, respectively, control the temperature and pressure. The velocity Verlet algorithm is used to integrate the equation of motion with a time step of 1 fs. For structures for which it is difficult to reach equilibration, the time step was reduced to 0.5 fs in the equilibration stage. After setting the temperature, another 10 ns of equilibration was performed using a microcanonical ensemble (NVE). Then, we recorded the heat

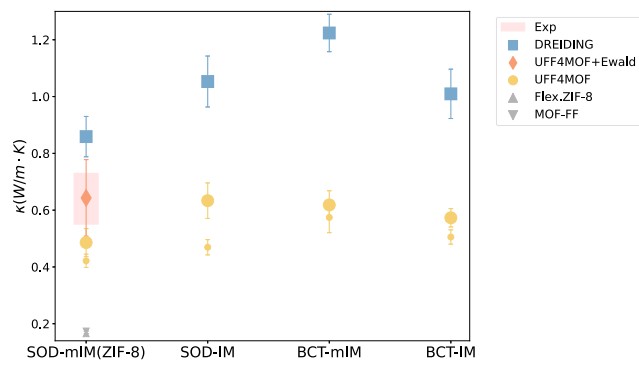

**Fig. 5 | Effect of the system size and force fields on the computed thermal conductivity.** The marker size represents the system size used in the simulation. Small markers correspond to 2 × 2 × 2 supercells, while large markers represent 4 × 4 × 4 supercells. Simulations using UFF4MOF with 4 × 4 × 4 supercells (0.49 ± 0.05 W m⁻¹ K⁻¹) show better agreement with experimental thermal conductivity (0.64 ± 0.09 W m⁻¹ K⁻¹ [41]) compared to those using 2 × 2 × 2 supercells and force fields such as MOF-FF and flexible ZIF-8 force field (both around 0.17 W m⁻¹ K⁻¹ [24,45]). Coulombic interactions are excluded from the simulations due to the high computational cost despite their potential to improve agreement with experiments.

flux in each direction within the structure for around 50 ns with an NVE ensemble simulation. The thermal conductivity in each direction was computed from the corresponding Green-Kubo relation[40]. An automatic fitting procedure to compute the converged thermal conductivity is provided in Section 2 in the SI.

We investigated the effects of system size and force fields (FF) on thermal conductivity in Fig. 5. We find that smaller sizes yield lower thermal conductivity due to finite-size effects. Simulations using UFF4MOF with 2 × 2 × 2 supercells (small yellow circles in Fig. 5) return lower thermal conductivity values than those using UFF4MOF with 4 × 4 × 4 supercells (big yellow circles). Among the ZIFs studied, SOD(mIM) corresponds to the experimental structure ZIF-8, for which the thermal conductivity of a single crystal has been experimentally measured at 0.64 ± 0.09 W m⁻¹ K⁻¹ (the red area in Fig. 5)[41]. For ZIF-8, simulations with 4 × 4 × 4 supercells produce results closer to the experimental value. We conducted a size effect study for all the ZIFs in our studies and ensured convergence to the bulk thermal conductivity.

We tested two force fields, DREIDING[42] and UFF4MOF[43,44], and compared them with other FFs reported in the literature. Our results show that the DREIDING force field (cyan squares) predicts an unusually high thermal conductivity. Simulation using other FFs, such as MOF-FF[24] and the flexible ZIF-8 force field[45] (gray triangles), yields even lower values than our UFF4MOF simulations with 2 × 2 × 2 supercells. While including Coulombic interactions in UFF4MOF (orange rhombus) improves consistency with experimental data, these interactions were excluded from our study due to their high computational cost. In addition, one advantage of UFF4MOF is its transferability, as it can be used for various MOFs.

We chose to use UFF4MOF and a sufficiently large system because the resulting combination strikes a good balance between accuracy in matching the experimental results and computational costs.

## Data availability
The CIF files and computed thermal conductivity data generated in this study have been deposited in the Zenodo database under accession code https://doi.org/10.5281/zenodo.14989563[31].

## Code availability
The Python code for generating the MATLAB circuit model scripts is available on GitHub (https://github.com/XiaoqZhang/CrystalCircuit) and archived in Zenodo as the supplementary software associated with the dataset https://doi.org/10.5281/zenodo.14989563[31].

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

## Acknowledgements

The Swiss National Science Foundation (SNSF) supports this work through an Advanced Grant (216165). The USorb-DAC Project, supported by a grant from The Grantham Foundation for the Protection of the Environment to RMI's climate tech accelerator program, Third Derivative, provided additional support. R.C. acknowledges PRACE for awarding us access to Piz Daint, at the Swiss National Supercomputing Centre (CSCS), Switzerland, with projects "pr128" and "s1019".

## Author contributions

X.Z. and R.C. performed the molecular dynamics simulations. S.B. developed the circuit model. Y.L. contributed to the discussions. B.S. and R.C. supervised the project. All authors reviewed, edited, and approved the manuscript.

## Funding

 Olavs Hospital - Trondheim University Hospital).

## Competing interests

The authors declare no competing interests.
