## [Transparent Peer Review file · Nature Communications]

Thermal Transport Mechanisms in ZIFs

Corresponding Author: Professor Raffaella Cabriolu

Version 0:

Reviewer comments:

Reviewer #1

(Remarks to the Author)

In this work, the authors developed a current circuit model to predict the thermal conductivity of 196 ZIF structures and a strong correlation between circuit model predicted and MD simulated thermal conductivity was revealed. In addition, it is also demonstrated that the impacts of functional groups on ZIFs are through the interplay between atomic mass and mechanical stability based on chosen ZIF structures. In general, this is an interesting work and the manuscript is well-written. However, some critical issues should be addressed before publication.

1. It is known that the thermal conductivity of MOFs/ZIFs are closely related with the crystal structure. What are the criteria of choosing the 196 ZIFs chosen for this study? How did the authors to guarantee the correctness of their ZIF structures. Since it has been reported previously that many structures in CORE_MOF have some problematic structures, e.g, some atoms of MOFs have unreasonable oxidation state numbers.
2. The current circuit model is the key of this study. However, the details on how to predict the thermal conductivity by such a model was not provided in the manuscript. Although Eq.1 is provided, but the detailed approach on how to obtain each parameter should be given in the main text.
3. According to Fig. 3, it is clear that the circuit model predicted thermal conductivity is generally higher than MD thermal conductivity. Is there any reason for such a phenomenon? Besides, for some ZIFs with high thermal conductivity above 0.5 W/mK, their thermal conductivity is significantly underestimated. Please explain on such a trend.
4. Compared with experimental measured thermal conductivity, which one is more accurate? Such discussion should be also supplemented. The universality of such current circuit model should be demonstrated by predicting some other MOFs with known experimental measured and MD simulated thermal conductivities.
5. Regarding the impacts of functional groups, the authors could provide the results from circuit model and MD simulation to validate the applicability of circuit model.
6. In Fig.4, the reduced thermal conductivity of dclM-based ZIF was attributed to the big mass difference from IM. However, nIM-based ZIF exhibited a bigger mass difference from IM, but their thermal conductivity is higher. In addition, the mechanical stability of such ZIFs should be also supplemented to demonstrate the interplay between atomic mass and mechanical stability.

(Remarks on code availability)

Reviewer #2

(Remarks to the Author)

This manuscript presented by Zhang and co-author explores the thermal transport mechanisms in ZIFs. Generally, understanding the insight influence of MOF structure on thermal properties is quite important for MOF design and thermal engineering. Herein, they develop a circuit model to calculate the reduced thermal circuit conductivity and reveal its correlation with thermal conductivity. I recommend this manuscript be published after clarifying the following issues.

1. As noted in the introduction, the developed parameter from limited structures illustrates the structure-property relationship hardly generalizable to a broader range of MOFs. How about the generalizable performance of such a circuit model applied

to other MOFs except these 196 ZIFs?

2. The heat resistance network is well-established in thermal engineering. The difference in such reduced thermal circuit conductivity is the introduced parameters of Li and Ai. What's the insight meaning of such parameters' influence on thermal conductivity? Without such parameters, how about the updated correlation results?

3. For the MOF crystal, the phonon is leading the thermal transfer, how about the functional group influence on phonon transport?

4. As provided in Figure S3, most ZIFs with the SOD and BCT net topologies exhibited relatively high bulk modulus (7-10 GPa), which is fairly close to the range of ZIFs(mIM) and ZIFs(nIM) (6-10 Gpa). So, why the BCT and SOD topology were except to obtain the Pearson coefficient. How about the overall Pearson correlation with such two topologies?

5. According to the definition, each chemical bond is modeled as a resistor with a 1Ω resistance. However, the influence of various chemical bonds (i.e., Zn-N, C-C) on thermal transport should be quite different. The thermal resistance of various chemical bonds can be explored to better illustrate the correlation. For example, the values for such 4 functional groups (mIM, nIM, IM, and dclM) can be determined.

(Remarks on code availability)

The code can be followed

Reviewer #3

(Remarks to the Author)

(Remarks on code availability)

Version 1:

Reviewer comments:

Reviewer #1

(Remarks to the Author)

The reply and revision have addressed all of my concerns. The current manuscript is acceptable.

(Remarks on code availability)

Reviewer #2

(Remarks to the Author)

We do think insight understanding the thermal conductivity of MOFs, COFs is quite important. Unlike the COFs, the various metal oxide nodes are quite important for determining the thermal conductivity and pose a challenge for the prediction of MOFs. After the revisions made by the authors, all my concerns have been addressed, and I can recommend this manuscript for publication in Nature Communications.

(Remarks on code availability)

Reviewer #3

(Remarks to the Author)

The authors have provided satisfactory responses to the reviewer comments. However, the generality of the proposed modeling approach remains a significant concern. For instance, the study only examines zeolites with thermal conductivities below $1 \text{ W m}^{-1} \text{ K}^{-1}$. It is unclear whether the model can be extended to zeolites or MOFs that exhibit higher thermal conductivities. For example, Islamov et al. (npj Computational Materials (2023) 9:11) have reported thermal conductivities exceeding $1 \text{ W m}^{-1} \text{ K}^{-1}$ for several MOFs. Can the authors' model accurately predict such values? If not, the applicability of the model appears to be quite limited.

(Remarks on code availability)

Response to the reviewers

Reviewer 1

Comments to the Author:

In this work, the authors developed a current circuit model to predict the thermal conductivity of 196 ZIF structures and a strong correlation between circuit model predicted and MD simulated thermal conductivity was revealed. In addition, it is also demonstrated that the impacts of functional groups on ZIFs are through the interplay between atomic mass and mechanical stability based on chosen ZIF structures. In general, this is an interesting work and the manuscript is well-written. However, some critical issues should be addressed before publication.

Reviewer Point P 1.1 — It is known that the thermal conductivity of MOFs/ZIFs are closely related with the crystal structure. What are the criteria of choosing the 196 ZIFs chosen for this study? How did the authors to guarantee the correctness of their ZIF structures. Since it has been reported previously that many structures in CoRE MOF have some problematic structures, e.g, some atoms of MOFs have unreasonable oxidation state numbers.

Reply: We used the same set as in a previous study. To clarify this, we added to the text:

The ZIFs studied in this work are the same set that was used by Moosavi et al. [1] to investigate the mechanical properties.

As an extra check of the correctness, we have run the new version of `mofchecker` and added to the revised manuscript:

To ensure the chemical consistency of all 196 ZIFs, we have run `mofchecker`[2] and no issues were detected.

Reviewer Point P 1.2 — The current circuit model is the key of this study. However, the details on how to predict the thermal conductivity by such a model was not provided in the manuscript. Although Eq.1 is provided, but the detailed approach on how to obtain each parameter should be given in the main text.

Reply:

We revised the main text, which now reads:

An example of a two-dimensional periodic crystal network is shown in Fig. 2a. In this representation, nodes correspond to atoms (or groups of atoms) located at the intersections of heat conduction paths, while edges denote the connecting atoms and chemical bonds. A temperature difference, e.g., T_1 on the left and T_2 on the right side of a unit cell, drives the heat flux. By simulating the resulting heat flow, the thermal conductivity of the crystal network can be determined.

To model this, we use an electricity analogy, mapping the crystal structure to a circuit. On the left boundary of the unit cell, the nodes with bonds crossing the left unit cell boundary are selected as starting points, and their equivalent nodes in the adjacent unit cell serve as endpoints.

These starting and endpoint nodes are treated as equipotential. Applying a voltage U across them creates a uniform electric field $E_i = U/L_i$, corresponding to a constant potential gradient along the unit cell direction L_i . The current density is given by $J_i = I/A_{\perp i}$, where $A_{\perp i}$ is the cross-sectional area perpendicular to the transport direction. Figure 2b shows the circuit corresponding to the 2D crystal network. In this illustrative case, with $U = 1\text{ V}$ and each resistor set to $R = 1\ \Omega$, the total current is $I = 15\text{ A}$. In the general three-dimensional case, Ohm's law yields the circuit conductivity as

$$\sigma_i = \frac{J_i}{E_i} = \frac{I/A_{\perp i}}{U/L_i}, i = x, y, z. \quad (1)$$

In the study of ZIFs, we construct the circuit model by defining the zinc

atoms as the network nodes. A resistor is placed between two nodes when they are connected by an organic linker, representing a heat conduction unit (Zn-linker-Zn, where the linker is IM, dcIM, mIM, or nIM), as shown in the inset of Fig. 2a. The corresponding circuit is generated using an in-house Python code[3].

Since the resistance of an individual heat conduction unit is not known a priori, we employ a reduced (dimensionless) formulation. Specifically, we set $U^* = 1$ and assign a resistance of unity to each heat conduction unit. The reduced current I^* is then obtained by circuit simulation in MATLAB Simulink[4]. The reduced circuit conductivity σ_i^* is calculated as

...

Reviewer Point P 1.3 — According to Fig. 3, it is clear that the circuit model predicted thermal conductivity is generally higher than MD thermal conductivity. Is there any reason for such a phenomenon? Besides, for some ZIFs with high thermal conductivity above $0.5 \text{ W m}^{-1} \text{ K}^{-1}$, their thermal conductivity is significantly underestimated. Please explain on such a trend.

Reply:

We analyzed the deviation distribution and included it in the SI:

Deviation of the thermal circuit prediction

Figure 1: The deviation distribution of the thermal circuit predictions. The deviation is defined as the difference between the thermal conductivity of the thermal circuit and the MD thermal conductivity.

In Fig. S8, we present the deviation of the thermal circuit predictions, defined as the difference between the thermal circuit conductivity τ and the MD thermal conductivity κ . Overall, the deviation distributions exhibit an approximately normal shape with only a slight rightward shift, indicating that the thermal circuit model generally provides unbiased predictions. However, a few outliers, such as the MD thermal conductivity values of BCT and SOD, exert a disproportionate influence on the fitted scaling factors τ_{linker} . These outliers enlarge the fitted values of τ_{linker} , thereby shifting the predicted conductivities upward. This effect is most pronounced in IM, where the distribution shows a stronger positive bias, while in dcIM the influence of such exceptional values is more moderate, leading to only a minor shift.

The structures for which the model underestimates the thermal conductivity correspond to two specific topologies, as discussed at the end of Section 2.3:

Two exceptions are observed in the network analysis: BCT in the z -direction and isotropic SOD in x -, y -, and z -directions, which can be attributed to their

relatively high bulk modulus[1] and similar zigzag heat conduction pathways (see Section 6 in the SI for more details).

Subsequently, further discussion is added:

In addition, these two topologies are characterized by the presence of 4-membered and 6-membered rings, structural motifs that tend to enhance the vibrational freedom of the metal centers.[5, 6] These additional vibrations may lead to variations in the thermal resistance of the corresponding heat conduction units.

Reviewer Point P 1.4 — Compared with experimental measured thermal conductivity, which one is more accurate? Such discussion should be also supplemented. The universality of such current circuit model should be demonstrated by predicting some other MOFs with known experimental measured and MD simulated thermal conductivities.

Reply: As to the generalisation of the circuit model to other MOFs, we added the following to the introduction to clarify why we focus on ZIFs:

In the introduction:

In MOFs, we typically have three parameters we can tune: the metal node, linker, and network topology. In this work, we focus on both the network topology and functionalization of the linker. Experimentally, for ZIFs only, we have been able to obtain a large number of ZIFs with the same metal node and linker but with different network topologies.[7]

...

In principle, similar studies can be done with MOFs, but there are far fewer experimental examples in which we can synthesize MOFs with the same linker and metal node but different pore topologies.[8]

To the best of our knowledge, there is only one experimental value available for the studied ZIFs, and we compared our simulations against this value. The good agreement indicates that our simulation provides a reliable prediction.

and in Discussion:

Although our work focuses on ZIFs, the proposed model provides a generalizable and efficient strategy that holds strong potential for extension to a broader range of MOFs, offering guidance for tailoring thermal conductivity through topology design and functionalization.

Reviewer Point P 1.5 — Regarding the impacts of functional groups, the authors could provide the results from circuit model and MD simulation to validate the applicability of circuit model.

Reply: The impacts of the functional groups on MD simulations and circuit model predictions follow the same trend, as shown in Fig. 3; we add the marginal distribution of both in Fig. 3 to clarify this:

Figure 2: The MD thermal conductivity κ versus the thermal circuit conductivity τ . Histograms of τ and κ , distinguished by linker type, are shown along the x- and y-axes, respectively. Thermal resistance coefficients are fitted for ZIFs with identical linkers, yielding good linear correlations when a single coefficient is applied. Pearson coefficients are calculated after excluding two exceptional net topologies, BCT and SOD, which are represented by squares.

We also added in the main text:

With these scaling factors, the thermal circuit conductivities (τ) of the four types of ZIFs converge onto a single universal linear correlation with the MD thermal conductivity.

Reviewer Point P 1.6 — In Fig.4, the reduced thermal conductivity of dcIM-based ZIF was attributed to the big mass difference from IM. However, nIM-based ZIF exhibited a bigger mass difference from IM, but their thermal conductivity is higher. In addition, the mechanical stability of such ZIFs should be also supplemented to demonstrate the interplay between atomic mass and mechanical stability.

Reply: dcIM linkers have a larger mass compared to nIM linkers. Note that in Fig. 1a, dcIM has two hydrogen atoms substituted by chlorine atoms.

Previous work has studied the mechanical stability of the ZIFs. We wrote in the main text:

Moosavi et al. [1] demonstrated that strategically placed functional groups can enhance mechanical stability by forming a secondary network of nonbonded interactions. Figure S3 in the SI, which is based on data from Moosavi et al. [1], shows that ZIFs with mIM and nIM linkers have a higher bulk modulus, and thus greater mechanical stability, compared to those with IM and dcIM linkers.

Reviewer 2

Comments to the Author: This manuscript presented by Zhang and co-author explores the thermal transport mechanisms in ZIFs. Generally, understanding the insight influence of MOF structure on thermal properties is quite important for MOF design and thermal engineering. Herein, they develop a circuit model to calculate the reduced thermal circuit conductivity and reveal its correlation with thermal conductivity. I recommend this manuscript be published after clarifying the following issues.

Reviewer Point P 2.1 — As noted in the introduction, the developed parameter from limited structures illustrates the structure-property relationship hardly generalizable to a broader range of MOFs. How about the generalizable performance of such a circuit model applied to other MOFs except these 196 ZIFs?

Reply: A similar comment was made by Reviewer 1, please see our reply P 1.4.

Reviewer Point P 2.2 — The heat resistance network is well-established in thermal engineering. The difference in such reduced thermal circuit conductivity is the introduced parameters of L_i and $A_{\perp i}$. What's the insight meaning of such parameters' influence on thermal conductivity? Without such parameters, how about the updated correlation results?

Reply: First, as mentioned in the main text, the circuit model has been utilized to investigate thermal transport in continuum materials. However, its application in simulating heat transfer within molecules remains unexplored, to the best of our knowledge.

In addition, L_i and $A_{\perp i}$ are not arbitrary fitting parameters but quantities that have physical meanings. Specifically, L_i determines the potential gradient (U/L_i) along the transport direction, and $A_{\perp i}$ provides the effective cross-section through which thermal current passes. Without these parameters, the model would reduce to a purely topological network. We have revised our manuscript to clarify this. Please find the details in our reply to Reviewer 1, point P 1.2.

Reviewer Point P 2.3 — For the MOF crystal, the phonon is leading the thermal transfer, how about the functional group influence on phonon transport?

Reply: Phonon transport in ZIFs has already been extensively investigated by Ying et al. [9], Zhou et al. [10]. We did not specifically study phonon transport, as our focus was on evaluating thermal conductivity using the Green-Kubo (GK) approach that inherently accounts for all contributions to heat transfer, including phonon transport. Therefore, we believe that the conclusions based on the GK results sufficiently capture the net impact of functional groups on thermal transport. We acknowledge that GK does not resolve phonon mechanisms; this would require mode-resolved analysis which maybe part of future work.

Reviewer Point P 2.4 — As provided in Figure S3, most ZIFs with the SOD and BCT net topologies exhibited relatively high bulk modulus (7-10 GPa), which is fairly close to the range of ZIFs(mIM) and ZIFs(nIM) (6-10 GPa). So, why the BCT and SOD topology were except to obtain the Pearson coefficient. How about the overall Pearson correlation with such two topologies?

Reply: Figure S3 shows that ZIFs with SOD and BCT net topologies exhibit relatively higher bulk moduli compared to other topologies with the same linker. For example, the red symbols representing SOD and BCT net topologies have a higher bulk modulus than the other red circles. As a result of their enhanced mechanical stability, these topologies also display higher MD thermal conductivity, as shown in Fig. 3.

We supplied the Pearson correlation coefficients calculated both including and excluding these two topologies in the SI:

In Figs. 3, S7a and S7b, we present the Pearson correlation coefficients calculated after excluding the two exceptional net topologies, BCT and SOD, which exhibit atypical thermal transport behavior. For comparison, Table S3 summarizes the Pearson correlation coefficients computed both including and excluding these two net topologies. Our analysis indicates that, given the large number of topologies studied, the fitted scaling factors remain rela-

tively robust to these outliers, whereas the Pearson correlation coefficients are more sensitive.

Linker	Including	Excluding
IM	0.67	0.69
dcIM	0.83	0.84
mIM	0.79	0.76
nIM	0.78	0.88
Fitted τ	0.86	0.92

Table 1: Comparison of Pearson correlation coefficients between thermal circuit predictions and MD thermal conductivity for different linkers, computed including and excluding the two exceptional net topologies, BCT and SOD.

Reviewer Point P 2.5 — According to the definition, each chemical bond is modeled as a resistor with a 1Ω resistance. However, the influence of various chemical bonds (i.e., Zn-N, C-C) on thermal transport should be quite different. The thermal resistance of various chemical bonds can be explored to better illustrate the correlation. For example, the values for such 4 functional groups (mIM, nIM, IM, and dcIM) can be determined.

Reply: We consider various chemical bonds as a group. Please see the details in our reply to point P 1.2 of Reviewer 1.

Reviewer 3

Comments to the Author: Reviewer #3 has not provided a full review but they have informed us that they have concerns about the novelty of your work as there are several related works on covalent-organic frameworks (COFs).

Reviewer Point P 3.1 — In particular the authors raise important concerns about how general the model is, and Reviewer #3 indicates that similar work has been reported for COFs. Indeed, it seems that further work, and a more comprehensive literature review are necessary to establish the significance of your results and the strength of conclusions you draw. Moreover, it is far from clear whether the work will continue to appear impressive in the light of this additional work. And although we would not rule out consideration of a revised manuscript, we feel that the present work is yet at too preliminary a stage to support publication in Nature Communications.

Reply: Reviewer #3 is correct that there is a large body of literature reporting experimental and computational studies of covalent-organic frameworks.

However, our study is a subclass of MOFs, specifically ZIFs, which are unique in that they mimic zeolite pore topologies. However, most importantly, this is a class of materials in which, similar to zeolites, one can experimentally change the network topology using the same metal node and linker. In addition, one can study the effect of functional groups. Hence, this provides a chemical design space with 49 distinct pore topologies and four different functional groups. **Such a subset of materials does not exist in COFs.** Hence, it is simply not possible to perform a similar study with COFs; therefore, we did not include references to the COF literature.

Most likely, Reviewer #3 missed this point that ZIFs do not have an analogy in COFs, but without a complete report, this is just speculation on our side.

In addition, the novelty of our work, in the context of what has been done in MOFs and COFs:

- **Circuit model** In the MOF literature, one can find screening studies that conclude

that thermal conductivity correlates with density. In this work, we demonstrate that a circuit model is a more natural and accurate approach to correlating thermal conductivity. Such a circuit model would also apply to COFs; however, upon reviewing the literature, we could not find any references to this approach. In fact, screening studies that are reported also use density as the main descriptor.

- **Effect of pore topology** in MOFs, for a given metal node and linker, one can have many different pore topologies. Currently, we only have insights into how to synthesize these different topologies for ZIFs, which are a special class of MOFs. In COFs, one could envision similar effects, but to the best of our knowledge, this is even less studied.
- **Effect of functional groups** This is something that can be studied systematically in COFs. However, we are not aware of any systematic study directly relevant to our research.

Our conclusion, after a careful analysis of the literature, is that the studies carried out were very similar to those in MOFs (except that many COFs are two-dimensional). Still, to the best of our knowledge, none of these studies were directly relevant to our work.

Because of these reasons, we did not refer to COFs in our article. In any case, we have added some general references in the introduction to the COF literature on thermal conductivity, to avoid the suggestion that we are not aware of the COF literature, which may have triggered the reaction of Reviewer #3.

References

- [1] Seyed Mohamad Moosavi, Peter G Boyd, Lev Sarkisov, and Berend Smit. Improving the mechanical stability of metal–organic frameworks using chemical caryatids. *ACS central science*, 4(7):832–839, 2018.
- [2] Xin Jin, Kevin Maik Jablonka, Elias Moubarak, Yutao Li, and Berend Smit. Mofchecker: a package for validating and correcting metal–organic framework (mof) structures. *Digital Discovery*, 2025.
- [3] Xiaoqi Zhang, Senja Barthel, Yutao Li, Berend Smit, and Raffaella Cabriolu. Unveiling thermal transport mechanisms in zifs: A circuit model approach to network and functional group design, March 2025. URL <https://doi.org/10.5281/zenodo.14989595>.
- [4] The MathWorks Inc. Matlab version: 24.2.0.2740171 (r2024b) update 1, 2024. URL <https://www.mathworks.com>.
- [5] Filip Formalik, Michael Fischer, Justyna Rogacka, Lucyna Firlej, and Bogdan Kuchta. Effect of low frequency phonons on structural properties of zifs with sod topology. *Microporous and Mesoporous Materials*, 304:109132, 2020.
- [6] Filip Formalik, Bartosz Mazur, Michael Fischer, Lucyna Firlej, and Bogdan Kuchta. Phonons and adsorption-induced deformations in zifs: is it really a gate opening? *The Journal of Physical Chemistry C*, 125(14):7999–8005, 2021.
- [7] Kyo Sung Park, Zheng Ni, Adrien P Côté, Jae Yong Choi, Rudan Huang, Fernando J Uribe-Romo, Hee K Chae, Michael O’Keeffe, and Omar M Yaghi. Exceptional chemical and thermal stability of zeolitic imidazolate frameworks. *Proceedings of the National Academy of Sciences*, 103(27):10186–10191, 2006.
- [8] Elizaveta I Yakovenko, Iurii M Nevolin, Anatoliy A Chasovskikh, Artem A Mitrofanov, and Vadim V Korolev. Data-driven prediction of structures of metal–organic frameworks. *Journal of Chemical Information and Modeling*, 65(4):1718–1723, 2025.

- [9] Penghua Ying, Jin Zhang, Xu Zhang, and Zheng Zhong. Impacts of functional group substitution and pressure on the thermal conductivity of zif-8. *The Journal of Physical Chemistry C*, 124(11):6274–6283, 2020.
- [10] Yanguang Zhou, Yixin Xu, Yufei Gao, and Sebastian Volz. Origin of the weakly temperature-dependent thermal conductivity in zif-4 and zif-62. *Physical Review Materials*, 6(1):015403, 2022.

Response to the reviewers

Reviewer 1

Remarks to the Authors:

The reply and revision have addressed all of my concerns. The current manuscript is acceptable.

Reply: Thank you for the comments and suggestions, which helped us improve the manuscript.

Reviewer 2

Remarks to the Authors:

We do think insight understanding the thermal conductivity of MOFs, COFs is quite important. Unlike the COFs, the various metal oxide nodes are quite important for determining the thermal conductivity and pose a challenge for the prediction of MOFs. After the revisions made by the authors, all my concerns have been addressed, and I can recommend this manuscript for publication in Nature Communications.

Reply: We sincerely thank the reviewer for their positive evaluation and supportive recommendation. We are pleased that the revisions have addressed all concerns, that we managed to clarify the role of metal oxide nodes and the challenges in predicting the ZIFs thermal conductivity.

Reviewer 3

Remarks to the Authors: The authors have provided satisfactory responses to the reviewer comments. However, the generality of the proposed modeling approach remains a significant concern. For instance, the study only examines zeolites with thermal conductivities below $1 \text{ W m}^{-1} \text{ K}^{-1}$. It is unclear whether the model can be extended to zeolites

or MOFs that exhibit higher thermal conductivities. For example, Islamov et al. (npj Computational Materials (2023) 9:11) have reported thermal conductivities exceeding $1 \text{ W m}^{-1} \text{ K}^{-1}$ for several MOFs. Can the authors' model accurately predict such values? If not, the applicability of the model appears to be quite limited.

Reply: We thank the reviewer for this comment. We would like to clarify that zeolites and ZIFs, although structurally related, are distinct classes of materials. Although zeolites and ZIFs share structural similarities, they differ significantly in bonding character and vibrational properties. ZIFs are hybrid organic–inorganic materials connected by metal–ligand coordination bonds, whereas zeolites are purely inorganic and covalently bonded. Therefore, the adaptation of the theoretical framework should take into account those important differences. The present study specifically focuses on ZIFs and zeolites were not within the scope of this work. We agree that applying the proposed model to zeolites, other MOFs, or porous materials with higher thermal conductivities would be an interesting direction for future research.